# TNFα-Induced LDL Cholesterol Accumulation Involve Elevated LDLR Cell Surface Levels and SR-B1 Downregulation in Human Arterial Endothelial Cells

**DOI:** 10.3390/ijms22126236

**Published:** 2021-06-09

**Authors:** Emmanuel Ugochukwu Okoro

**Affiliations:** Department of Microbiology, Immunology and Physiology, Meharry Medical College, Nashville, TN 37208, USA; eokoro@mmc.edu

**Keywords:** cell surface receptor, cholesterol, endothelial cell, lipoprotein, low-density lipoprotein (LDL), tumor necrosis factor (TNF)

## Abstract

Excess lipid droplets are frequently observed in arterial endothelial cells at sites of advanced atherosclerotic plaques. Here, the role of tumor necrosis factor alpha (TNFα) in modulating the low-density lipoprotein (LDL) content in confluent primary human aortic endothelial cells (pHAECs) was investigated. TNFα promoted an up to 2 folds increase in cellular cholesterol, which was resistant to ACAT inhibition. The cholesterol increase was associated with increased ^125^I-LDL surface binding. Using the non-hydrolysable label, Dil, TNFα could induce a massive increase in Dil-LDL by over 200 folds. The elevated intracellular Dil-LDL was blocked with excess unlabeled LDL and PCSK9, but not oxidized LDL (oxLDL), or apolipoprotein (apoE) depletion. Moreover, the TNFα-induced increase of LDL-derived lipids was elevated through lysosome inhibition. Using specific LDLR antibody, the Dil-LDL accumulation was reduced by over 99%. The effects of TNFα included an LDLR cell surface increase of 138%, and very large increases in ICAM-1 total and surface proteins, respectively. In contrast, that of scavenger receptor B1 (SR-B1) was reduced. Additionally, LDLR antibody bound rapidly in TNFα-treated cells by about 30 folds, inducing a migrating shift in the LDLR protein. The effect of TNFα on Dil-LDL accumulation was inhibited by the antioxidant tetramethythiourea (TMTU) dose-dependently, but not by inhibitors against NF-κB, stress kinases, ASK1, JNK, p38, or apoptosis caspases. Grown on Transwell inserts, TNFα did not enhance apical to basolateral LDL cholesterol or Dil release. It is concluded that TNFα promotes LDLR functions through combined increase at the cell surface and SR-B1 downregulation.

## 1. Introduction

TNFα was originally described as a tumor-selective hemorrhagic factor [1,2]. However, it is now recognized to have multiple cell-type dependent effects [3,4]. Released as a 17 kD soluble protein from a transmembrane precursor by tumor necrosis factor converting enzyme (TACE) [5,6,7], this cytokine mainly promotes changes that facilitate leukocyte (inflammatory) functions [3,8]. Some of its early intracellular cascade molecules include the rapid-acting transcription factor, nuclear factor kappa B (NF-κB), which induces the expression of other pro-inflammatory mediators such as interleukin 1 and 2 [9,10]. Likewise, TNFα is capable of inducing apoptosis [11,12,13], possibly through c-jun N-terminal kinase (JNK) activation [14,15,16]. Thus, TNFα may clear a way for the passage of leukocytes through target epithelial cell apoptosis. In contrast, it can directly or indirectly induce the proliferation of lymphocytes, fibroblasts, and endothelial cells [12,17,18,19,20]. This ability to induce proliferation has been reported to be due to the upregulation of growth factor receptors [20,21] or growth factor release, such as vascular endothelial growth factor (VEGF) [20,22,23].

The pro-atherogenic arm of TNFα is in part due to its ability to enhance monocyte entry into the intima through induction of adhesion molecules on arterial endothelial cells. In atherosclerotic lesions, macrophages [22] and smooth muscle cells [22,23] have been found to express TNFα, where it acts on overlying endothelial cells to stimulate the expression of leukocyte adhesion molecules such as E-selectin, vascular cell adhesion molecule-1 (VCAM-1), and intercellular adhesion molecule-1 (ICAM-1) [24]. Moreover, it induces expression of monocyte chemoattractant protein I (MCP-1) [25], which stimulates the gradient migration of lymphocytes and monocytes [26,27].

The serum concentration of TNFα rises with atherosclerotic severity [28]. Thus, it has been reported to be about 200 folds higher in the atherosclerotic lesion than in the circulating blood [29]. Moreover, age [30], vascular infection [31,32], and psycho-environmental stress [33] are associated with elevated serum TNFα. Although it is considered a pro-inflammatory cytokine, the effect of TNFα on atherogenesis is surprisingly mixed. For example, tumor necrosis factor receptor 1 (TNFR1) has been reported to be anti-atherogenic [34,35], pro-atherogenic [36,37,38], or neutral [39]. Likewise, TNFR2 has been reported to be pro-atherogenic [40] or neutral [35]. On one hand, it has been reported to promote advanced lesion development [36,41,42,43]. However, others have reported that it suppresses the early stages of the disease [43], while antibodies against TNFα have been shown to be atherogenic [44].

Unlike macrophages [45,46] and smooth muscle cells [45,47], confluent endothelial cells, which represent the form in which they exist in the arterial wall, are resistant to cholesterol accumulation [48]. Specifically, in the atherosclerotic lesion, macrophages and smooth muscle cells become cholesteryl ester laden cells called foam cells, in association with oxidized LDL (oxLDL) uptake via scavenger receptor A (SR-A) [49]. Macrophages naturally express large quantities of SR-A, whereas smooth muscle cells can be induced to express it [50]. However, atherosclerotic lesions also contain endothelial foam cells within an intact monolayer [51,52,53]. The expression of lectin-like oxidized LDL receptor 1 (LOX-1) has been reported to be inducible in endothelial cells [54]. This may contribute to endothelial foam cell formation. However, oxLDL induces endothelial apoptosis [55,56]. Thus, the knowledge that atherosclerotic endothelial foam can exist without apparent damage [53] raises the question as to whether the concentration of oxLDL is sufficient to induce endothelial foam cell formation in vivo.

In this report, the effect of TNFα on the pro-atherogenic lipoprotein, LDL [57,58] is investigated. Evidence is presented that native LDL uptake alteration in pHAECs may contribute to the atherogenic nature of TNFα and LDL. It is shown that TNFα increases pHAEC cholesterol and lipid accumulation by enhancing LDL receptor (LDLR) functions.

## 2. Results

### 2.1. TNFα Enhances Cholesterol Accumulation and LDL Binding to pHAECs

Atherosclerotic lesion is characterized by release of inflammatory cytokines, among which is TNFα. Prior to the initiation of experiments, the pHAECs obtained from ATCC were routinely verified through cellular morphological criteria and VWF expression [59]. As shown in Figure 1A, the pHAECs expressed VWF. The characteristic “cobble stone” appearance of endothelial cells can be seen at the level of confluency shown in Figure 1B. Prolonged maintenance at confluence increases the cell number, while the individual cell size is decreased (Figure 1C). The in vivo level of arterial endothelial confluency is closer to that seen in Figure 1C [53]. To evaluate the effect of TNFα on pHAECs cholesterol content, confluent pHAECs were treated with or without TNFα and the cellular cholesterol content was measured. As is shown in Figure 1D, TNFα significantly increased unesterified cholesterol, and to a greater degree, esterified cholesterol (Figure 1E). To evaluate whether TNFα affects the ability of LDL to bind to the cells, ^125^I-LDL binding was performed. Figure 1F shows that TNFα enhanced ^125^I-LDL surface releasable ^125^I-LDL at 4 °C, after the cells were pre-treated at 37 °C. The amount released increased in the presence of 20 folds unlabeled LDL. This indicates that the ^125^I-LDL bound to some native LDL receptor. Further, cell-associated ^125^I-LDL was also higher on TNFα-treated pHAECs (Figure 1G).

It has been reported that TNFα induces cholesteryl ester accumulation in monocytes through enhanced ACAT activity in the presence of oxLDL [60]. To evaluate the extent to which enhanced ACAT activity is responsible in raising pHAEC cholesterol in the presence of LDL, the cells were treated with or without ACAT inhibitor, Sandoz 58-035. As reported in Figure 2A,B, the inhibition of ACAT activity slightly raised the unesterified cholesterol, and significantly suppressed cholesteryl ester accumulation. Despite this inhibition, the TNFα-induced total cholesterol accumulation, now mainly in the unesterified form, was not prevented (Figure 2A,C). ACAT inhibition also reduced the total cholesterol in the cells, with or without TNFα stimulation (Figure 2C). This is a known effect of ACAT inhibition. The higher elevated unesterified cholesterol is more likely to be effluxed from the cells than the esterified one [61].

### 2.2. ACAT Inhibitor Does Not Prevent TNFα-Induced LDL Cholesterol Accumulation

It has been reported that TNFα induces cholesteryl ester accumulation in monocytes through enhanced ACAT activity in the presence of oxLDL [60]. To evaluate the extent to which enhanced ACAT activity is responsible in raising pHAEC cholesterol in the presence of LDL, the cells were treated with or without ACAT inhibitor, Sandoz 58-035. As is reported in Figure 2A,B, inhibition of ACAT activity slightly raised the unesterified cholesterol, and significantly suppressed cholesteryl ester accumulation. Despite this inhibition, the TNFα-induced total cholesterol accumulation, now mainly in the unesterified form, was not prevented (Figure 2A,C). ACAT inhibition also reduced the total cholesterol in the cells, with or without TNFα stimulation (Figure 2C). This is a known effect of ACAT inhibition. The higher elevated unesterified cholesterol is more likely to be effluxed from the cells than the esterified one [61].

### 2.3. LDL Oxidation Is Not Required for TNFα-Induced LDL Accumulation

Having demonstrated that TNFα promoted LDL binding to pHAECs (Figure 1), the requirement for oxidative modification of LDL was investigated. Particularly, TNFα has been reported to promote release of the reactive oxygen species, superoxide and hydrogen peroxide [62,63]. Hence, experiments were performed to determine whether an oxidative modification of LDL is a prerequisite to TNFα-induced LDL binding. To visualize TNFα-induced LDL binding and subsequent internalization, Dil-LDL was used. Internalized Dil-LDL was significantly increased through TNFα pre-treatment (Figure 3A,B). It can also be seen in Figure 3A,B that excess unlabeled native LDL blocked the binding and internalization of Dil-LDL, with and without TNFα treatment. This demonstrates that native LDL components, and therefore receptors, are required for Dil-LDL binding and internalization. Excess unlabeled oxLDL, on the other hand, did not prevent Dil-LDL intracellular accumulation in control and TNFα-treated cells. Instead, there was a tendency of oxLDL to enhance intracellular Dil-LDL with or without TNFα.

### 2.4. TNFα Induces Massive Dil over [^3^H]CE Lipid Accumulation from LDL

The lipids, Dil and ^3^H-cholesteryl esters ([^3^H]CE), are stably fixed within LDL. To evaluate the ability of pHAECs to retain LDL hydrolysable [^3^H]CE or the non-hydrolysable Dil, the cells were treated with increasing concentration of TNFα in the presence of [^3^H]CE-LDL or Dil-LDL. As shown in Figure 4A, ^3^H-cholesterol accumulation derived from [^3^H]CE-LDL was increased by about 2 folds with increasing TNFα concentration. On the other hand, Dil accumulation increased by about 50 folds (Figure 4B). Next, the cells were treated with increasing Dil-LDL concentration without (Figure 4C) or with TNFα (Figure 4D). As is shown in Figure 4C, 400 µg/mL Dil-LDL increased intracellular Dil level by about 50 folds compared to that at 1 µg/mL under control condition. Compared to Ctrl at 400 µg/mL Dil-LDL, TNFα increased the intracellular Dil by over 200 folds (Figure 4D).

### 2.5. Lysosomal Inhibitor Enhances TNFα-Induced LDL Lipid Accumulation

The lowering of pH facilitates the disintegration and degradation of internalized LDL by low pH-dependent lysosomal enzymes. This degradation is suppressed by the pH-raising compound, chloroquine [64]. To evaluate whether a similar phenomenon occurs in pHAECs, the cells were treated in the presence or absence of chloroquine. The presence of chloroquine caused the Dil to accumulate circumferentially within the cells, presumably in defective lysosomes (Figure 5A). As can be seen in Figure 5A,B, chloroquine greatly enhanced Dil-LDL accumulation without TNFα. The presence of TNFα further increased the amount of chloroquine-induced cellular Dil-LDL. Chloroquine also increased ^3^H-cholesterol level from [^3^H]CE-LDL. However, it was not as pronounced as in the case of Dil (Figure 5C).

### 2.6. ApoE Is Not Required for TNFα-Induced Dil-LDL Accumulation

Apolipoprotein B (apoB) is the major protein in LDL [65]. However, variable amounts of apoE are also present in LDL [66]. Since apoE plays a major role in binding several lipoprotein receptors of the LDLR family [67,68], the effect of apoE depletion on TNFα-induced Dil-LDL intracellular levels was investigated. The successful depletion of apoE from Dil-HDL3 and Dil-LDL can be seen in Figure 6A. That both HDL3 and LDL contain apoE, while apoA1 and apoB are present in significant amounts only in HDL3 and LDL, respectively, is apparent in the figure. Data in Figure 6A also show that apoE represented a small proportion of the total LDL particle, but a larger fraction in HDL3. The apoB protein is nearly equal with or without apoE depletion, whereas apoA1 is noticeable less. That equal protein was loaded is presented in the Appendix A. In Figure 6B, it can be seen that the isolation procedure did not inhibit the ability of TNFα to induce Dil-LDL accumulation. Further, apoE depletion had no effect in this regard.

### 2.7. TNFα-Induced Dil-LDL Accumulation Is Blocked by Specific LDLR Antibody

Having found evidence that TNFα promotes cholesterol accumulation in pHAECs through the endocytosis of native LDL, I next investigated what members of the LDLR family are responsible. As can be seen in Figure 7A, the pan-LDLR family blocker PCSK9 [69,70,71], but not RAP [72], suppressed the Dil-LDL accumulation. To investigate the contribution of LDLR in mediating TNFα-induced Dil-LDL accumulation, an LDLR antibody was investigated for its specificity and function in pHAECs treated with or without TNFα. As can be seen in Figure 7B, the antibody recognized a single protein consistent with the expected migration position of LDLR. It can be seen that incubation of TNFα-treated cells with the antibody induced an upward migratory shift in LDLR protein to a larger extent than in control cells. Attempts to directly detect the LDLR antibody were unsuccessful, suggesting that the shift is due to another interaction or modification.

HDL3 critically depends on non-LDLR mechanisms for uptake, especially in the absence of apoE [73]. Thus, to further evaluate the specificity of the LDLR antibody, intracellular uptake of Dil-HDL3 was measured. As can be seen in Figure 7C, intracellular Dil-HDL3 levels were significantly blocked by excess unlabeled HDL3, whereas that of LDL only had a mild effect. Further, the LDLR antibody did not block intracellular Dil-HDL3 to a considerable level. In contrast, the LDLR antibody blocked intracellular Dil-LDL accumulation (Figure 7D). This blockage was greater than 99% in TNFα-treated cells.

### 2.8. TNFα Upregulates Cell Surface LDLR Protein

Having determined that LDLR is responsible for TNFα-induced LDL accumulation in pHAECs, I next tested whether the total and surface LDLR is changed by TNFα. As can be seen in Figure 8A,D, TNFα-induced an upregulation of the LDLR protein total and surface by about 90% and 138%, respectively. ABCA1 (Figure 8A,B) and ICAM-1 (Figure 8A,C) proteins, both of which have been reported to be induced by TNFα treatment [32,74,75], were found to be significantly increased as well. A low molecular weight form of ABCA1, however, was primarily induced in TNFα-treated cells. In contrast, scavenger receptor B1 (SR-B1) was downregulated by TNFα (Figure 8A,E). Similar findings have been reported on hepatocytes [76]. Figure 8A also shows that the intracellular proteins GAPDH were undetectable at the cell surface. Although caveolin-1 has been reported to be closely associated with the inner plasma membrane [77,78], it was not detected in the cell surface assay. This indicates that the outer plasma membrane proteins were selectively labeled.

### 2.9. TNFα Promotes Rapid Association of LDLR with Its Antibody

To evaluate the consequence of the increase in cell surface LDLR induced by TNFα on the association and entry of LDLR antibody, immunofluorescence studies were performed. As can be seen in Figure 9A, TNFα did not enhance the association of control antibody with the cells, but significantly induced association with the specific LDLR antibody by 30 min at 37 °C. This indicates that TNFα did not promote non-selective endocytic processes. The surface distribution of LDLR (arrows) and the internalization of the surface LDLR (arrow heads) in cells treated with TNFα can be seen in Figure 9B.

### 2.10. Antioxidant Suppresses TNFα-Induced Dil-LDL Accumulation

TNFα signaling cascade involves multiple mediators, among which are the transcription factor, NF-κB, and apoptosis signal-regulating kinase 1 (ASK1), an upstream mediator of JNK and p38 [79], that facilitate apoptosis. To evaluate whether these proteins are upstream in the TNFα-induced Dil-LDL accumulation, experiments were performed in the presence of their respective inhibitors. As can be seen in Figure 10A, the inhibitors did not suppress the action of TNFα. In contrast, the antioxidant, TMTU, inhibited this effect with increasing doses (Figure 10B).

### 2.11. TNFα Does Not Affect AP to BL Release of Degraded LDL Protein

To address the effect of increased LDLR-mediated uptake of LDL on apical to basolateral LDL protein transport, the pHAECs were grown to confluence on Transwell inserts, as shown in Figure 11A. ^125^I-LDL detected in the BL medium was either non-intact (Figure 11B) or intact (Figure 11C). In Figure 11B, it is shown that unlabeled LDL competitor reduced the amount of non-intact ^125^I-LDL, indicating release in a receptor-dependent manner. This was TNFα-independent. In contrast, the unlabeled competitor had no effect on the intact ^125^I-LDL measured in the BL medium (Figure 11C). TNFα tended to enhance BL release of intact ^125^I-LDL, consistent with enhanced cell-cell permeability. The overall data thus indicate that the LDLR protein cargo is not efficiently trafficked in a polarized manner in pHAECs.

### 2.12. TNFα Does Not Affect AP to BL LDL Lipid Release

To determine whether LDLR facilitates AP to BL LDL lipid transport, the pHAECs on Transwell inserts were incubated with LDL colabeled with Dil and ^3^H-cholesteryl esters (Figure 12A,B). As can be seen in the figures, TNFα dose-dependently increased the accumulation of Dil-LDL on the pHAECs grown on Transwell inserts, in the same manner as those grown on plastic dishes. Similar to pHAECs on plastic dishes, the LDLR antibody significantly blocked the TNFα-induced uptake of Dil-LDL by pHAECs grown on Transwell inserts by over 99%. The corresponding uptake of ^3^H-cholesteryl esters is shown in Figure 12C. The ^3^H-cholesteryl ester accumulation induced by TNFα is blocked in the presence of antibody against LDLR. Surprisingly, LDLR blockage had no effect on the amount of measurable ^3^H-choleseryl ester detectable in the BL medium (Figure 12D). This is consistent with the BL Dil-LDL, whose value was independent of TNFα treatment (Figure 12E).

## 3. Discussion

The modification of LDL, producing derivatives with receptors present or inducible on macrophages or smooth muscle cells drives foam cell formation. Thus, LDL modification due to oxidation [80], aggregation [81], or complex formation with proteoglycans [82,83,84,85] is known promote macrophage foam cells. The present results demonstrate that native LDL can itself induce excess lipid accumulation in endothelial cells, independent of modification, in the presence of TNFα. The data suggest that pHAECs, like macrophages, may be susceptible to changes related to excess cholesterol loading and may facilitate the progression of atherosclerotic lesion in the continued presence of elevated LDL or apoB-containing lipoproteins, which enters cells through the LDLR.

Excess cholesterol or lipids in arterial endothelial may to lead to endothelial dysfunction due, in part, to space and membrane disruption [86]. The reduction of serum LDL cholesterol to or below 100 mg/dl has been shown to reduce the negative consequences of atherosclerosis [87]. The general upregulation of the LDLR receptor function through inhibition of PCSK9 has been reported to benefit against atherosclerotic coronary artery disease [88,89]. Notwithstanding, PCSK9 also downregulates other receptors in the LDLR family [70,71], and binds the scavenger receptor, CD36, concurrent with thrombosis [88,90]. The benefit of targeting PCSK9 to upregulate LDLR in the liver, thereby reducing serum LDL, is obvious for those with atherosclerotic cardiovascular diseases for which other strategies are ineffective. My results suggest, however, that additional tissue specificity may be required, particularly if therapy fails to reduce serum non-HDL lipoproteins below optimal levels. Specifically, elevated non-HDL lipoprotein concentration in the setting of elevated LDLR may disrupt the normal function of arterial endothelial cells, and perhaps other cells.

The data from this report indicate that pHAECs have efficient mechanism(s) of getting rid of LDL-derived cholesterol. TNFα induced about 25 folds intracellular Dil-LDL compared to [^3^H]CE-LDL. Thus, it appears that pHAECs are normally protected against LDL-derived cholesterol accumulation by keeping cell surface LDLR down, combined with efficient efflux mechanisms. The TNFα-induction of ABCA1 (Figure 8), which mediates cholesterol efflux, suggest that ABCA1 may be involved. However, the significance of an apparently truncated form (Figure 8A) is unclear. The tendency of TNFα to induce ABCA1 expression in some cell types has also been published by others. Specifically, it has been reported that TNFα enhances cholesterol efflux and ABCA1 mRNA and protein through NF-kB signaling [75] in macrophages, and possibly adipocytes [91]. However, as is often characteristic of the complex nature of TNFα signaling, it has also been reported to downregulate ABCA1 in hepatocytes [92], Caco-2 intestinal cell [93], and osteocytes [94].

TNFα is a pleiotropic autocrine and paracrine mediator important in multiple signaling cascades that range from activating immune cells to fight viruses, bacteria, and cancer cells, to promoting entry of monocytes and lymphocytes into atherosclerotic lesions [4,95,96]. The latter process is recognized to be pro-atherogenic. TNFα has also been reported to play an important role in the current COVID-19 pandemic [97]. Thus, nonspecific disruption of TNFα activity in the animal interferes with many important biological functions [98]. It has also been reported that TNFα upregulates LDLR in hepatocytes [92]. My finding that TNFα upregulates surface LDLR function in pHAECs implies that it is possible to regulate this process selectively, while leaving the beneficial functions of TNFα intact. 

It has been reported that AP to BL LDL transport occurs in vivo across arterial endothelial cells [99]. The findings from this report suggest that this transport is probably not through LDLR, as lysosome inhibition (Figure 5) significantly promoted Dil-LDL accumulation with or without TNFα. SR-BI, on the other hand, has been reported to faciliate AP to BL transport of LDL in an LDLR-independent manner [100]. However, there is some controversy about the role of LDLR in trafficking ligand across the brain endothelium. Thus, it has been reported that LDLR mediates transcytosis of LDL and its ligand in brain microvascular endothelial cells [101] in vitro. However, studies using animal models suggest that LRP and very low-density lipoprotein receptor (VLDLR), not LDLR, are responsible for the transcytosis [102]. 

In summary, the actions of TNFα on LDL uptake in pHAECs are enhanced through the cell surface upregulation of LDLR, combined with the downregulation of SR-B1. Because SR-B1 is also a receptor for LDL, its downregulation promotes preferential binding and internalization through LDLR. Unlike SR-B1 which does not promote net LDL cholesterol accumulation in cells [102], LDLR is a potent inducer of LDL-derived cholesterol storage due to trafficking of its cargo to lysosomes. The finding that the inhibition of NF-κB or apoptosis mediators did not significantly affect TNFα-induced Dil-LDL accumulation suggest that alternate signaling cascades are involved.

## 4. Materials and Methods

### 4.1. Materials

The list of items used can be found under Appendix A.

### 4.2. Cell Culture and Incubations

pHAECs, ATCC, lots 70001318, 63233442, and 64323512, passages 3–7, were used during the experiments, with similar results. The majority of experiments were performed using lot 70001318. Lot details are available on the ATCC website. The characteristics were routinely verified by criteria of cobblestone morphology at confluence and expression of the von Willenbrand factor. The cells were maintained in VBM with growth factor kit (VEGF) in the presence of 15% FBS, and antibiotic, antimycotic (100 units/mL of penicillin, 100 µg/mL of streptomycin, and 0.25 µg/mL of amphotericin B. Under serum-free conditions, the cells were cultured in VBM, VEGF, with the latter antimicrobials and 0.1% FAF-BSA. Incubations were at 37 °C, unless indicated otherwise. The experiments were best started with pHAECs confluent for 5 days or longer. Separate cells from individuals (2–36 years old) were used during the course of the experiments, with similar results. Incubations were 24 ± 5 h. For cell-associated studies at 37 °C, the cells were washed 3× with ice cold mHBSS. Imaging studies demonstrated that the vast majority of Dil-lipoproteins were within the cell under these conditions. For intracellular LDL levels, pHAECs were incubated on ice for 5 min with ice cold 400 units/mL sodium heparin in mHBSS, washed twice with the same buffer, then two more times with ice cold mHBSS, to remove surface LDL. In experiments using excess unlabeled lipoproteins, the volume represented 3% or less of the culture medium. Lipids were extracted as described below or the cells were fixed with 4% paraformaldehyde in PBS prior to microscopy.

### 4.3. Lipoprotein Purification

All procedures were performed between on ice to 4 °C. Centrifugations and dialysis were performed at 4 °C. Lipoproteins were isolated from freshly drawn human blood (BioIVT) anticoagulated with Na_2_EDTA. Consenting and ethical standard procedures were conducted by the company. The blood was centrifuged at 4000× *g* for 30 min to obtain the plasma. To the plasma, butylated hydroxytoluene (BHT) in DMSO was added to 45 µM (0.01%). Then, sequential density ultracentrifugations [103] were performed to obtain d < 1.006 (VLDL), d = 1.019–1.063 (LDL), or d = 1.12–1.21 (HDL3) g/mL. Ultracentrifugations were done using type Ti70 rotor at 50,000 rpm for 20 (VLDL and LDL) or 48 h (HDL3). The lipoproteins were dialyzed through ~4 kD molecular weight cut-off membrane in 3 successions against ~180 times dialysis buffer (DB): 10 mM Tris-HCl, 150 mM NaCl, 0.3 mM Na_2_EDTA, pH 7.5, with deionized water, in the dark, each lasting about 24 h. After concentration using 3 kD MWCO centrifugal filters, the LDL was filtered through 0.2 µm membrane under sterile conditions. Lipoprotein contents were routinely verified by western blotting and coomassie staining. In addition, the cholesterol and cholesteryl ester contents were verified to be consistent with previous publications [104,105,106]. Lipoprotein concentrations represent the protein content throughout the manuscript.

### 4.4. Cellular Cholesterol Determination

The lipids were extracted into hexane: isopropanol (1:1) at room temperature for 3 h, then the solvent was evaporated at room temperature using centrivap concentrator (Labconco). The extracted lipids were redissolved in isopropanol. To determine the unesterified cholesterol content, the lipids in isopropanol were mixed with 10 times volumes of cholesterol assay buffer (CAB): 0.1% FAF-BSA, 2 mM sodium taurocholate, 50 mM Tris-HCl, pH 7.5, 0.3 mM Na_2_EDTA, 5% isopropanol, and 250 mM sucrose, with freshly added 0.5 U/mL horseradish peroxidase (HRP) and 0.02 U/mL cholesterol oxidase in the presence of 30–50 µM scopoletin [107] (excitation 360, emission 460). The CAB was kept at room temperature to ensure full solubilization of the lipids. This enzymatic approach is a modification of a previously published procedure [108]. Total cholesterol was determined as above, with the addition of 0.1 U/mL cholesteryl esterase. Readings were obtained after incubation at 37 °C for 20 min. Cholesteryl esters were determined by subtracting unesterified cholesterol from the total cholesterol. The residual cell matter after lipid extraction was lysed with 0.1 M NaOH, 0.1% SDS, for protein reading using BCA assay kit.

### 4.5. Transwell Insert Experiments

pHAEC (about 0.5 × 10^5^ cells/mL) were seeded to ~ 80% confluence on 3 µm pore membrane inserts without coating and allowed to reach confluence for 5 days or longer, as judged from fluorescence imaging, and increasing resistance to [^3^H]-inulin transport. The cell culture conditions were as indicated above. Arterial endothelial cells have been reported to form their own basement membrane with continued culture [109]. This approach represents a physiologically relevant extracellular matrix for the pHAECs. Cellular attachment to the inserts was achieved through limited trypsinization of pHAECs on plastic dishes. In brief, cells on plastic dishes were covered with trypsin-EDTA solution for 30 s, then removed. Following incubation at 37 °C for about 3 min, a further trypsinization of the detached cells was quenched by resuspending the cells in complete culture medium as described in Section 2.2. The suspension was transferred to the inserts, containing the same medium on the basolateral side. Typically, when the cells reached full confluency, media on the apical side failed to leak to the basolateral compartment after overnight incubation at 37 °C. Details of the transport experiments are as described under Figure 11 and 12 legends. During the transport experiments, serum-free medium on the basolateral side was added such that the apical and basolateral heights were the same.

### 4.6. ^3^H-Cholesteryl Ester ([^3^H]CE) Generation

First, 40 µCi of ^3^H-cholesterol in ethanol was added in 10 µL aliquots to 40 mL of 0.2 µm filtered human serum under sterile conditions. The mixture was incubated at 37 °C for 48 h in the dark. ^3^H-cholesteryl esters and other lipids were extracted into chloroform: methanol [110]. After evaporation of the solvent, the residue was dissolved in a minimal volume of chloroform. Subsequently, ^3^H-cholesteryl esters were purified using serial bond elut columns with several hexane passages, as previously described [111].

### 4.7. Western Blotting

pHAECs were lysed with lysis buffer (150 mM NaCl, 2% Triton X-100, 20 mM HEPES, pH 7.4, 2% protease inhibitor and 2% phosphatase inhibitor by sonication at 2 setting (Fisher sonic dismembrator model 100) on ice for 10 s. After centrifugation at 16,000× *g*/10 min at 4 °C, equal volume of loading buffer (8 M urea, 2% SDS, 125 mM Tris-HCl, pH 7.0, 5% glycerol, 10% beta-mercaptoethanol, 0.06% bromphenol blue) was mixed with the supernatant at room temperature. This was followed by electrophoresis, electrical transfer to PVDF membrane, and immunoblotting.

### 4.8. Dil- and/or [^3^H]CE-Lipoproteins

The generation of Dil-labeled lipoproteins was obtained essentially as previously described [112], with minor modifications. In brief, a mixture of about 2 mg HDL3, LDL, or VLDL protein and 7 mL of human lipoprotein deficient serum was mixed with ~1 mg [^3^H]CE and/or 0.5 mg Dil in 10 µL DMSO aliquots under sterile conditions. After covering with foil, the mixture was incubated at 37 °C for 21 h. The lipoproteins were repurified as described above, after adjusting to their respective densities with solid KBr. Combined Dil, [^3^H]CE-LDL radioactivity was 8.1 dpm/µg protein.

### 4.9. LDL Iodination with Na^125^I

About 2 mg LDL protein in PBS and 10 iodination beads were incubated with Na^125^I for 10 min at room temperature in glass vials. The transformation was stopped with 50 mM each of unlabeled sodium iodide and sulfite, and 100 µM BHT in DMSO (0.01%). The labeled LDL was washed with desalting columns, then passed through centrifugal filters, 3 kD MWCO, against dialysis buffer.

### 4.10. ^125^I-LDL Cell Surface Binding

Confluent pHAECs on plastic dishes were cultured with 0 or 5 ng/mL TNFα in 15% serum for 48 h. The treatment continued with serum-free medium in the continued presence of TNFα for 3 h to deplete surface-bound LDL. Subsequently, ^125^I-LDL was added to 15 µg/mL for 1 h at 37 °C. The cells were then washed two times with PBS at room temperature, then chilled on ice. Afterwards, serum-free medium at 4 °C with 0 (buffer) or 300 µg/mL LDL was added. Following additional incubation at 4 °C for 1 h, the radioactivity released to the medium was taken as surface-releasable ^125^I-LDL.

### 4.11. LDL Oxidation

LDL was oxidized essentially as described [113]. Briefly, LDL was dialyzed against PBS at 4 °C to remove antioxidants, passed through 0.2 µm filter, and incubated at 37 °C for 22 h with 5 µM CuSO_4_ in PBS under sterile conditions. After adding BHT in DMSO (0.005%) to 1 µM and Na_2_EDTA to 10 mM, the mixture was dialyzed against dialysis buffer (under Lipoprotein purification), and finally filtered through 0.2 µm membrane under sterile conditions.

### 4.12. TBARS Assay

A 2-thiobarbituric assay to estimate the extent of LDL oxidation was determined essentially as previously described [114], with some modifications. oxLDL was mixed successively with 3 volumes of 0.67% 2-thiobarbituric acid in 50 mM NaOH and 20% trichloroacetic acid, each containing 1 mM Na_2_EDTA, respectively. After heating at 55 °C for 1 h, the mixture was centrifuged at 16,000× *g*/30 s. Fluorescence (530 nm excitation, 590 nm emission) reading in the supernatant was determined. Malonaldehyde bis-(dimethyl acetal) was used as a malondialdehyde (MDA) precursor for determining malondialdehyde standards. Care should be taken when performing this procedure, as volatile products are generated at 100 °C. Standards up to 5 nmol/μL were used. oxLDL was measured at 54 nmol MDA equivalents/mg protein.

### 4.13. Depletion of apoE from Dil-Lipoproteins

Normal mouse IgG agarose or apoE3 agarose beads were washed 4× with dialysis buffer under sterile conditions. Subsequently, Dil-HDL3 or Dil-LDL was added to the beads in dialysis buffer. Following 24 h incubation at 4 °C, the unbound lipoproteins were sterile filtered through 0.2 µm membranes.

### 4.14. Cell Surface Biotinylation

Serum media from confluent pHAECs were replaced with 400 U/mL sodium heparin in mHBSS on ice for 5 min, washed 2× with the the same buffer, then 1× with ice cold mHBSS. After 1× wash with ice cold 1× PBS, the cells were treated with EZ Link Sulfo NHS Biotin in 1× PBS for 30 min at 4 °C. Unreacted biotin reagent was quenched with 100 mM glycine in PBS at 4 °C for 5 min. Following 1× wash with ice cold mHBSS, the cells were lysed on the plates with lysis buffer as indicated above under Western blotting on ice for 30 min, followed by sonication at 2 setting for 10 s on ice. Aliquots of the supernatant was mixed with 1 volume of 2× urea loading buffer without heating. The rest of the supernatant was mixed with strepavidin mag sepharose (pre-washed 2× with incubation buffer: 1% BSA, 0.1% NaN3, 0.1% Triton X-100). After 10× dilution with the incubation buffer containing 0.5% each of protease inhibitor and phosphate inhibitor cocktails, the suspension was swirled for 12 h at 4 °C. The beads were washed 4× with precipitation buffer, then mHBSS at 4 °C. After 1× urea loading buffer (1:1 volume of lysis buffer and 2× urea loading buffer), with 1% protease and phosphatase inhibitors, respectively, the samples were heated at 95 °C for 5 min to solubilize the bound biotinylated proteins.

### 4.15. Dextran-Mn Separation of Intact and Non-Intact ^125^I-LDL

Intact ^125^I-LDL was separated from non-intact LDL using dextran sulfate, Mn^2+^ procedure essentially as previously described [115]. To the BL medium, d < 1.21 g/mL FBS was added to 15% to produce a visible precipitate, mixed, then dextran sulfate and MnCl were added to 65 mg/mL and 0.2 M, respectively. After incubation for 20 min at room temperature, the mixtures were centrifuged at 5000× *g*/5 min at 4 °C. The pellet was redissolved with 20 mg/mL dextran sulfate before scintillation counting.

### 4.16. Immunofluorescence

Confluent pHAECs in serum medium were incubated with 0 or 100 ng/mL TNFα for 24 h. Subsequently, the media were replaced with serum-free medium containing 10 µg/mL normal goat IgG or LDLR Ab (R&D Systems) in the continued presence of 0 or 100 ng/mL TNFα for 0, 5, 30, or 120 min at 37 °C. Afterwards, the cells were chilled on ice and further incubated at 4 °C for 1 h. Then the cells were washed 2× with ice-cold mHBSS, fixed with ice-cold methanol at −20 °C for 10 min, washed once with mHBSS at room temperature, then blocked with 50% human serum in mHBSS, 0.1% sodium azide (blocking buffer) for 1 h at room temperature. Afterwards, it was replaced with 4 µg/mL red antigoat antibody in blocking buffer. Following 30 min incubation at room temperature, the medium was replaced with mHBSS, washed 3× with blocking buffer (10 min each at room temperature), fixed with 4% paraformaldehyde in PBS, followed by fluorescence microscopy.

### 4.17. Fluorescence Microscopy

Fluorescence microscopy was performed using Keyence phase-contrast fluorescence microscope. Images were analyzed using CellProfiler [116]. Confocal microscopy was performed using glass bottom dishes (Cellvis).

### 4.18. Statistical Analysis

Data are reported as averages ± standard deviation. *n* = number of independent experiments performed on separate days. Most experiments were performed with an average of 3 or greater replicates. Analysis of variance, followed by Tukey post-hoc testing was done by using statpages.info website.

## Figures and Tables

**Figure 1 ijms-22-06236-f001:**
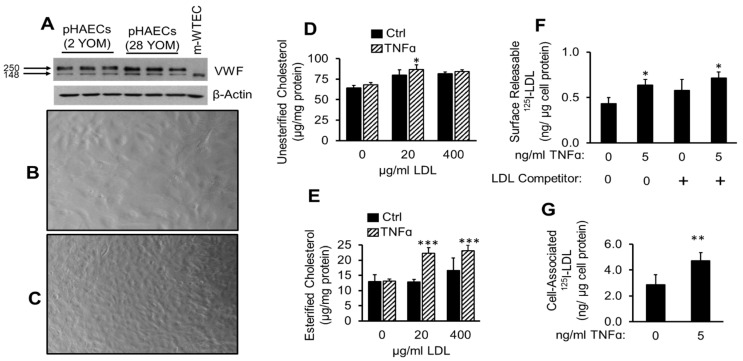
TNFα enhances cholesterol accumulation and LDL binding to pHAECs. (**A**) Expression of von Willebrand factor (VWF) by 2- or 28-year-old male (YOM) pHAECs, lots 70001318, 63233442, respectively. Mouse wild-type endothelial cells (m-WTEC) were used a comparison. (**B**,**C**) ranges of pHAEC confluency used. Images are at 20× magnification. (**D**–**E**) pHAECs were treated with the indicted concentration of human LDL without (Ctrl) or with 100 ng/mL TNFα for 24 h. Unesterified cholesterol and cholesteryl esters were determined as described under Section 2. *n* = 5. (**F**) Cells were pre-treated with 0 or 5 ng/mL TNFα in serum medium as described under Section 2. After incubation at 37 °C, the amount of ^125^I-LDL released to serum-free medium at 4 °C without (0) or with 20 folds unlabeled LDL competitor (+) was determined as the surface ^125^I-LDL. (**G**) The experiment was performed as in (**E**), and the amount of radioactivity associated with the cell was determined. (**C**,**D**) *n* = 3. *, **, ***, *p <* 0.05, 0.01, 0.001, relative to the corresponding Ctrl.3.2. ACAT inhibitor does not prevent TNFα-induced LDL cholesterol accumulation.

**Figure 2 ijms-22-06236-f002:**
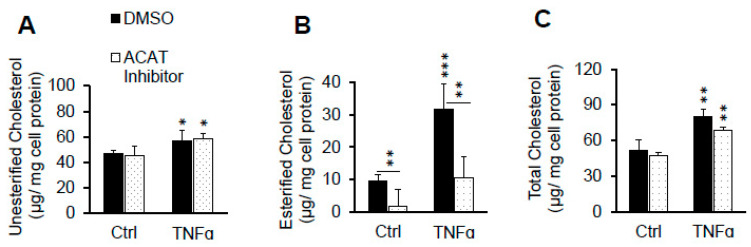
ACAT inhibitor does not prevent TNFα-induced LDL cholesterol accumulation. Cells in serum-free medium were treated with 0 or 100 ng/mL TNFα in the presence of 0.1% DMSO (DMSO) or 10 µg/mL ACAT inhibitor (Sandoz 58-035) and 100 µg/mL LDL for 24 hrs. After washing with heparin as described under Materials and Methods, the cellular cholesterol content was determined. (**A**) unesterified cholesterol, (**B**) esterified cholesterol, and (**C)** total cholesterol. *n* = 3. *, **, ***, *p* < 0.05, 0.01, 0.001, relative to corresponding Ctrl or as indicated with bars.

**Figure 3 ijms-22-06236-f003:**
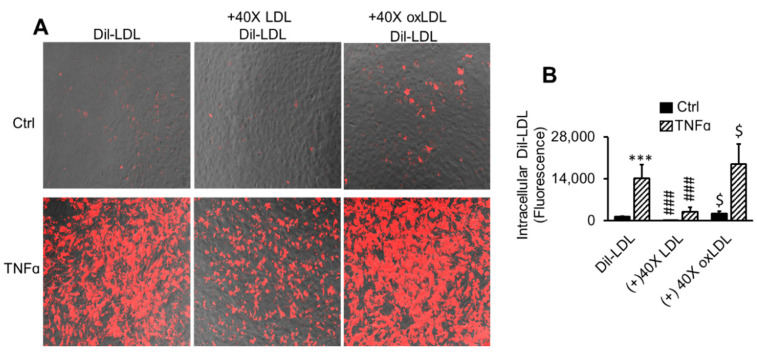
LDL oxidation is not required for TNFα-induced LDL accumulation. (**A**,**B**) Cells were treated with 0 (Ctrl) or 100 ng/mL TNFα (TNFα) for 24 h in serum medium. The media were replaced with serum-free media in the continued presence of TNFα for 2 h to remove surface lipoproteins. Finally, 5 µg/mL Dil-LDL in serum-free medium (Dil-LDL) ± 40 times LDL (+40× LDL) or oxLDL (+40× oxLDL) was added, followed by 3 h incubation. The images are 20× magnifications. *n* = 3. ***, *p* < 0.001 vs. Ctrl. $, ###, *p* < 0.05, 0.001, relative to Dil-LDL alone.

**Figure 4 ijms-22-06236-f004:**
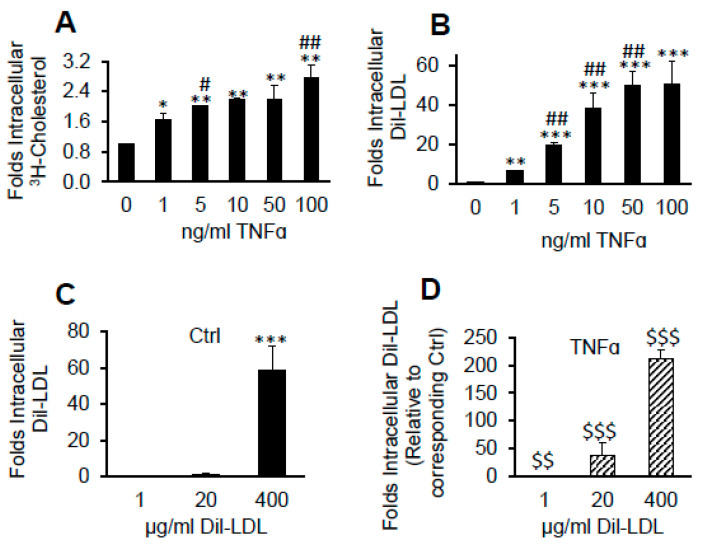
TNFα induces massive Dil over [^3^H]CE lipid accumulation from LDL. pHAECs in serum medium were pre-treated with the indicated concentration of human TNFα for 24 h. Subsequently, the cells were treated in the continued presence of the previous TNFα concentration and 50 µg/mL [^3^H]CE-LDL (**A**) or 5 µg/mL Dil-LDL (**B**) without serum for 24 h. A, *n* = 4, B, *n* = 3. The cells were treated without (Ctrl), (**C**), or with 100 ng/mL TNFα (**D**), as above, in the presence of increasing concentration of Dil-LDL. *n* = 5. *, **, ***, *p <* 0.05, 0.01, 0.001, relative to 0 ng/mL TNFα or 1 µg/mL Dil-LDL. #, ##, *p <* 0.05, 0.01, vs. previous TNFα dose. $$, $$$, *p* < 0.01, 0.001, relative to the corresponding Dil-LDL concentration under Ctrl condition.

**Figure 5 ijms-22-06236-f005:**
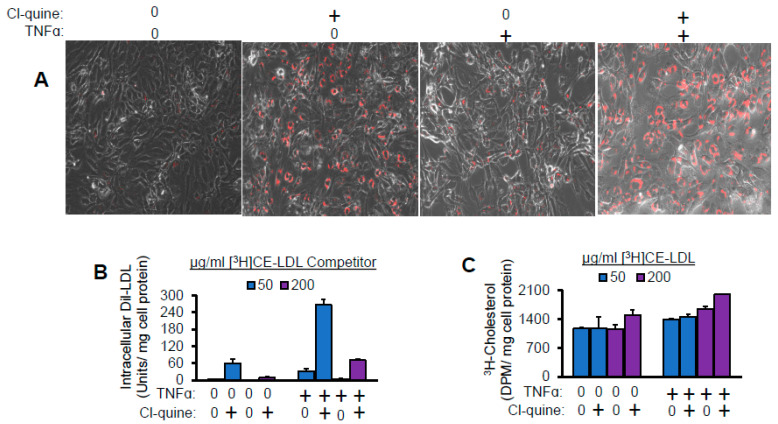
Lysosomal inhibitor enhances TNFα-induced LDL lipid accumulation. The cells were pre-treated with 0 or 100 ng/mL TNFα (+) for 24 h in the presence of serum. Subsequently, in the continued presence of TNFα, the cells were treated with 0.5 µg/mL Dil-LDL, plus 50 or 200 µg/mL [^3^H]CE-LDL, and containing 0 or 50 µM chloroquine diphosphate (Cl-quine), +, a lysosomal blocker, without serum. After 24 h, the intracellular fluorescence (**A**,**B**) or radioactivity (**C**) was determined. The separation of #cells in A is an artificact of preparation prior to lysing the cells for radioactivity. The circumferential accumulation of the Dil-LDL in the presence of chloroquine is apparent (**A**). The images are 20× magnifications. *n* = 2.

**Figure 6 ijms-22-06236-f006:**
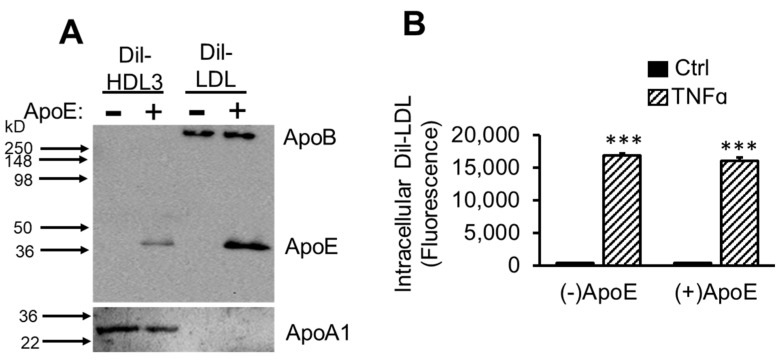
ApoE is not required for TNFα-induced Dil-LDL accumulation. (**A**) Dil-HDL3 or Dil-LDL was depleted of apoE (−), or apoE was retained (+) as described under Section 2. The presence of apoB and apoE were determined by immunoblotting 0.5 and 1 µg/mL Dil-HDL3 and Dil-LDL, respectively. (**B**) pHAECs were treated ± 100 ng/mL TNFα in serum medium for 24 h. Following 2 h incubation in serum free medium, in the continued presence of TNFα, the cells were incubated with 2.5 µg/mL Dil-LDL without (−) or with apoE (+) for 3 h. *n* = 3. ***, *p* < 0.001, relative to Ctrl.

**Figure 7 ijms-22-06236-f007:**
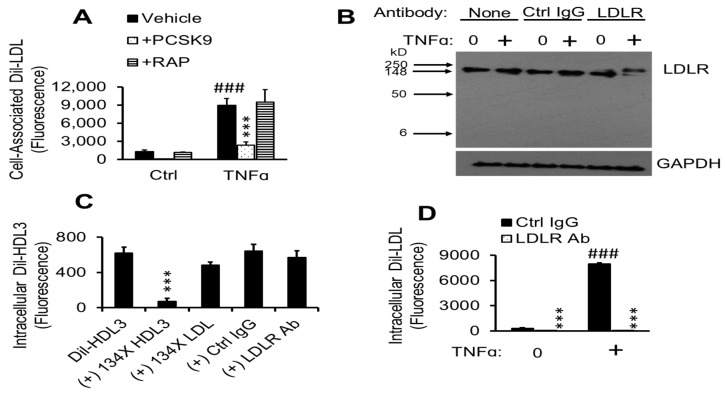
TNFα-induced Dil-LDL accumulation is blocked by specific LDLR antibody. (**A**) pHAECs in serum medium were pre-treated with 0 (Ctrl) or 100 ng/mL TNFα for 24 h. Afterwards, the cells were cultured in the presence of vehicle, 12.5 µg/mL human PCSK9 (ApoER, LDLR, LRP, and VLDLR inhibitors), or 21 µg/mL human RAP (ApoER2, LRP, and VLDLR inhibitors) in serum-free medium for 1 h. Lastly, 8 µg/mL Dil-LDL was added, followed by 4 h incubation. *n* = 3. ###, ***, *p* < 0.001 vs. Ctrl vehicle and TNFα vehicle, respectively. (**B**), Cells were cultured in serum medium ± 100 ng/mL TNFα, followed by 3 h incubation in serum-free medium in the continued presence of TNFα (+) without antibody (None), 18 µg/mL control IgG (Ctrl Ig) or LDLR antibody (LDLR), then immunoblotted for LDLR and GAPDH. *n* = 3. (**C**), pHAECs were cultured in serum-free medium for 2 h, followed by incubation with 5 µg/mL Dil-HDL3 with (+) 134× unlabeled HDL3, LDL, 20 µg/mL control IgG (Ctrl IgG) or LDLR antibody (LDLR Ab), as indicated. *n* = 6. ***, *p* < 0.001 vs. Dil-HDL3. (**D**) Cells were cultured in serum medium with 0 or 100 ng/mL TNFα (+), followed by 2 h incubation in serum-free medium in the continued presence of TNFα. Subsequently, 5 µg/mL Dil-LDL in the presence of 20 µg/mL control IgG (Ctrl IgG) or LDLR antibody (LDLR Ab) was added. Note that the values for LDLR Ab are very low compared to others. *n* = 8. ###, ***, *p* < 0.001 vs. corresponding 0 µg/mL TNFα and Ctrl IgG, respectively.

**Figure 8 ijms-22-06236-f008:**
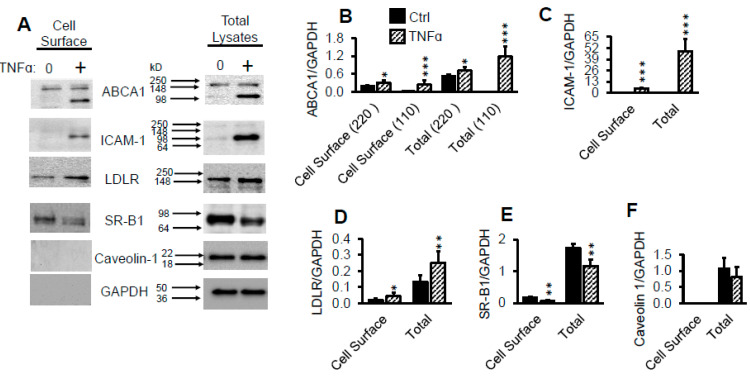
TNFα upregulates cell surface LDLR protein. (**A**–**F**), Confluent pHAECs in serum medium were pre-treated with 0 or 100 ng/mL TNFα (+) for 24 h in serum medium. The cells were then treated and lysed for total (Total Lysates) and biotinylated cell surface (Cell Surface) proteins, as explained under Materials and Methods, Section 4.14. ABCA1 bands (**A**,**B**) at ~220 and 110 kD are indicated. *n* = 3. *, **, ***, *p <* 0.05, 0.01, 0.001, vs. corresponding Ctrl.

**Figure 9 ijms-22-06236-f009:**
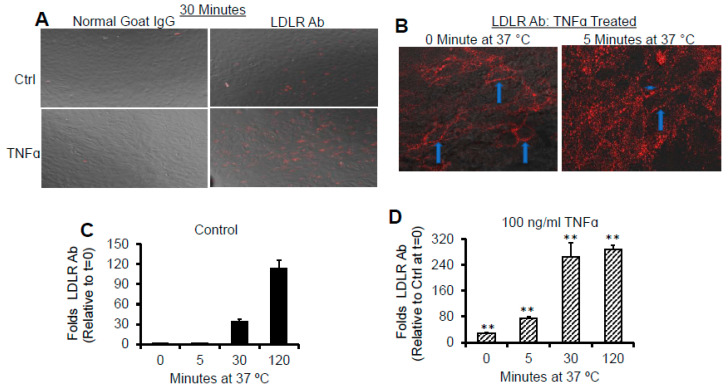
TNFα promotes rapid association of LDLR with it antibody. pHAECs in serum medium were pre-treated with 0 (Ctrl) or 100 ng/mL TNFα (TNFα) for 24 h. Subsequently, in the continued presence of TNFα, the media were replaced with 5 µg/mL control normal goat IgG or LDLR antibody (LDLR Ab) for 0, 5, 30, or 120 min at 37 °C. The surface-accessible LDLR was then determined as detailed under Experimental Procedures. (**A**) 20× objective magnification of LDLR detected after 30 min. (**B**) 40× confocal detection of LDLR in TNFα-treated pHAECs at 0 and 5 min. The arrows show the membrane distribution of the LDLR in focus, while the arrowhead shows internalized LDLR. (**C**) Surface-accessible LDLR as a function of time under control condition. (**D**) As C, in the presence of TNFα. *n* = 4. **, *p <* 0.01 relative to control at the same time.

**Figure 10 ijms-22-06236-f010:**
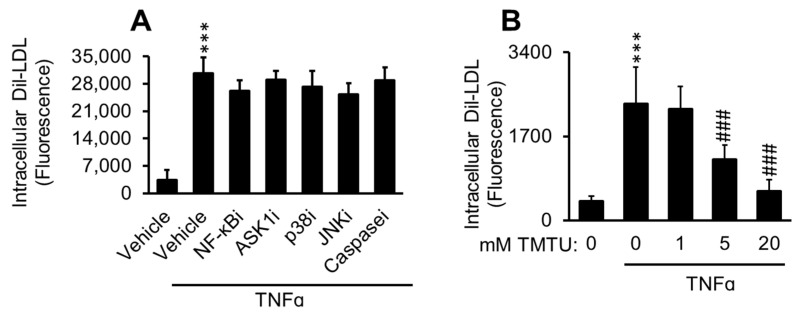
Antioxidant suppresses TNFα-induced Dil-LDL accumulation. (**A**) pHAECs were pre-treated with 0.05% DMSO (Vehicle) or TNFα-cascade inhihitors, as indicated: NF-kBi (25 µM). Two different forms were tested. ASK1i (5 µM), p38i (1 µM), JNKi (5 µM), or pan-caspase inhibitor, Caspasei (5 µM) for 30 min in serum-free medium. Subsequently, serum-free 3 µg/mL Dil-LDL, with 100 ng/mL TNFα, as indicated, was added, followed by 24 h incubation. *n* = 3. (**B**) Cells were treated with 1% DMSO vehicle with 100 ng/mL TNFα as indicated, in the presence of 0, 1, 5, or 20 mM TMTU and 3 µg/mL Dil-LDL. Subsequently, the cells were cultured for 24 h. *n* = 4. ***, ###, *p <* 0.001 vs. 0 µg/mL TNFα and 0 mM TMTU in the presence of TNFα, respectively.

**Figure 11 ijms-22-06236-f011:**
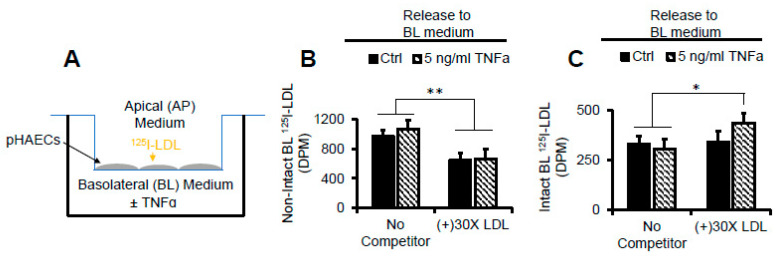
TNFα does not affect AP to BL release of degraded LDL protein. (**A**) Diagram of the experimental setup in (**B**,**C**) in which pHAECs were grown to confluence on 3 µm (pore size) polycarbonate filters, with ^125^I-LDL in the AP medium. (**B**) pHAECs in serum medium were pre-treated with 0 (Ctrl) or 5 ng/mL TNFα on the BL side for 48 h as described under Materials and Methods, Transwell Insert Experiments. Under serum-free conditions, 10 µg/mL ^125^I-LDL was added to the AP side (No Competitor) or with 30 folds unlabeled LDL, (+) 30× LDL, in the continued presence of TNFα. After 3 h, the BL media were collected, and non-intact ^125^I-LDL was measured as described under Experimental Procedures. (**C**) The experiment was performed as in B, followed by detection of intact BL ^125^I-LDL. *n* = 3. *,**, *p <* 0.05, 0.01, respectively.

**Figure 12 ijms-22-06236-f012:**
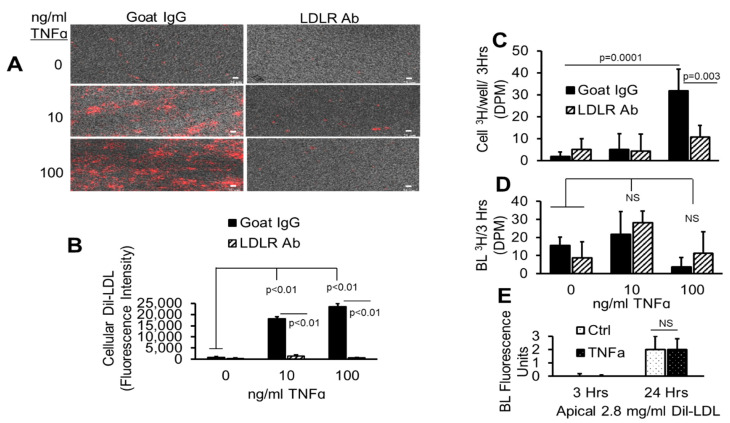
TNFα does not affect AP to BL LDL lipid release. (**A**–**D**) pHAECs were pre-treated as in Figure 11 with 0, 10, or 100 ng/mL TNFα for 24 h in serum. Subsequently, 20 µg/mL control normal goat IgG (Goat IgG) or LDLR antibody (LDLR Ab) was added to the AP medium without serum for 2 h. Finally, 1.6 mg/mL double labeled Dil, [^3^H]CE-LDL was added to the AP medium for 3 h. After heparin wash, the intracellular fluorescence (**A**,**B**), ^3^H radioactivity (**C**), or medium BL ^3^H (**D**) was determined. (**E**) The cells were treated for 3 or 24 h with 0 (Ctrl) or 100 ng/mL TNFα without serum in the presence of apical Dil-LDL, and the BL fluorescence was determined. *n* = 3.

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
