# Peer review of "TNFα-Induced LDL Cholesterol Accumulation Involve Elevated LDLR Cell Surface Levels and SR-B1 Downregulation in Human Arterial Endothelial Cells"

_ijms, 2021, doi:10.3390/ijms22126236_

Round 1
Reviewer 1 Report
In this original paper, author have utilized immortalized pHAECs from ATCC to carry various in vitro experiments and highlighted that the TNFɑ-induced LDL cholesterol accumulation involve elevated LDLR cell surface levels and SR-B1 downregulation. Most of the previously published paper focused on the accumulation oxidized LDL inside macrophages through scavenging receptor and chnage its phenotype to foam cells (accumulation of fatty streak) and subsequent release of various inflammatroy mediators leadinf to progression of vascular inflammation/atheroscelrosis. The results presented in this study is very interesting and manuscript is very well written. This article falls under the scope of IJMS. I would like to recommend author to address few issues before considering appropriate for publication
Comments
- Why did author choose to perform experiment in confluent pHAEC? Noseda et al., (PMID 15456857) showed that when endothelial cells reach confluence, Notch is activated and p21Cip1 is downregulated. Inhibition of the Notch pathway at confluence prevents p21Cip1 downregulation and induces Rb phosphorylation. Endothelial cells undergo senescence if they are too confluent and chronic exposure of TNF-α to endothelial cells leads to premature senescence (PMID 28045034). So, it is likely that the activity/function of pHAEC differ in confluent and sub-confluent state. Please provide rationale/justification.
- Introduction: line 44: Background inflammation is an independent risk factor for atherosclerotic cardiovascular disease. This sentence is not very clear. Does author mean vascular inflammation induced by TNF-alpha by particular cell types (example: cytokines release by vascular smooth muscle cell or macrophage)?
- Introduction: line 50-21 “In contrast, it but suppresses early fatty streak” Again this is not clear. Fatty streak of what? Foam cell (macrophage) or other cell type (endothelial). Please be specific
- Introduction: According to the title of the manuscript, the information provided for endothelial cell is sufficient. However, I felt that the information about vascular smooth muscle cells and macrophage are not enough. Their role in atherosclerosis progression is very important. Please elaborate their role together with scavenger receptor.
- Section 2.2, line 93-94 “For cell-associated studies at 37 ºC, cells were washed 3X with mHBSS at 0 ºC”. Please give rationale for washing cell at 0 degree C. This temperature seems to be quiet odd as mHBSS get frozen or nearly frozen at 0 degree. Does author mean ice cold mHBSS?
- Section 2.2, Line 104, Author used human blood for lipoprotein isolation. Please include the details of human ethics approval for the use of human blood.
- Section 2.10. Please cite the method of LDL oxidation with CuSO4.
- Figure 1A and 1B, TNF-alpha is treated to pHAECs at 100ng/ml and for figure 1C and 1D, TNF-alpha is treated at 5ng/ml. Any reason for this difference in dose of TNF-alpha
- Figure 7B, the experiment on mouse hepatocytes is done n=1. For scientific publication, reproducibility and statistical analysis, author need to present at least n=3 data.
- Texts are cryptic in many places. Needs extensive grammatical correction and spelling error correction.
- Line 461, what does 3mm of trans well insert mean? Is that a pore size? Please elaborate
- Line 425, TNF-a is repeated, please delete. Line 430, please replace “nN” with “n”
Author Response
Reviewer 1
In this original paper, author have utilized immortalized pHAECs from ATCC to carry various in vitro experiments and highlighted that the TNFɑ-induced LDL cholesterol accumulation involve elevated LDLR cell surface levels and SR-B1 downregulation. Most of the previously published paper focused on the accumulation oxidized LDL inside macrophages through scavenging receptor and chnage its phenotype to foam cells (accumulation of fatty streak) and subsequent release of various inflammatroy mediators leadinf to progression of vascular inflammation/atheroscelrosis. The results presented in this study is very interesting and manuscript is very well written. This article falls under the scope of IJMS. I would like to recommend author to address few issues before considering appropriate for publication
Comments
- Why did author choose to perform experiment in confluent pHAEC? Noseda et al., (PMID 15456857) showed that when endothelial cells reach confluence, Notch is activated and p21Cip1 is downregulated. Inhibition of the Notch pathway at confluence prevents p21Cip1 downregulation and induces Rb phosphorylation. Endothelial cells undergo senescence if they are too confluent and chronic exposure of TNF-α to endothelial cells leads to premature senescence (PMID 28045034). So, it is likely that the activity/function of pHAEC differ in confluent and sub-confluent state. Please provide rationale/justification.
Response. Arterial endothelial cells exist in vivo in a confluent state [1]. “Full” confluency was chosen to simulate this condition. While TNFɑ has the capacity to cause endothelial cell cycle arrest and/or apoptosis, the pHAECs were resistant to this effect in the presence of serum and VEGF included in the media, within the 24-48 hour interval of treatment. This is illustrated in Figures 3 and 9. In vivo, the endothelial monolayer remains intact in uncomplicated atherosclerotic lesions, despite production of TNFɑ by macrophages and smooth muscle cells [1-5].
- Introduction: line 44: Background inflammation is an independent risk factor for atherosclerotic cardiovascular disease. This sentence is not very clear. Does author mean vascular inflammation induced by TNF-alpha by particular cell types (example: cytokines release by vascular smooth muscle cell or macrophage)?
Response. Thanks for pointing this out. The sentence has been removed for clarity.
- Introduction: line 50-21 “In contrast, it but suppresses early fatty streak” Again this is not clear. Fatty streak of what? Foam cell (macrophage) or other cell type (endothelial). Please be specific
Response. It has been reworded.
- Introduction: According to the title of the manuscript, the information provided for endothelial cell is sufficient. However, I felt that the information about vascular smooth muscle cells and macrophage are not enough. Their role in atherosclerosis progression is very important. Please elaborate their role together with scavenger receptor.
Response. It has been elaborated.
- Section 2.2, line 93-94 “For cell-associated studies at 37 ºC, cells were washed 3X with mHBSS at 0 ºC”. Please give rationale for washing cell at 0 degree C. This temperature seems to be quiet odd as mHBSS get frozen or nearly frozen at 0 degree. Does author mean ice cold mHBSS?
Response. “ice cold” was intended. This has been changed.
- Section 2.2, Line 104, Author used human blood for lipoprotein isolation. Please include the details of human ethics approval for the use of human blood.
Response. The blood was obtained from the company, BioIVT, which conducts ethical and consenting practices. The information has now been included.
- Section 2.10. Please cite the method of LDL oxidation with CuSO4.
Response. It has been cited.
- Figure 1A and 1B, TNF-alpha is treated to pHAECs at 100ng/ml and for figure 1C and 1D, TNF-alpha is treated at 5ng/ml. Any reason for this difference in dose of TNF-alpha
Response. The lower concentration was used to demonstrate that binding differences are detectable at this concentration. Figure 4 shows dose-curve responses.
- Figure 7B, the experiment on mouse hepatocytes is done n=1. For scientific publication, reproducibility and statistical analysis, author need to present at least n=3 data.
Response. The data has been removed and added as a supporting information, instead.
- Texts are cryptic in many places. Needs extensive grammatical correction and spelling error correction.
Response. Additional error checks have been done.
- Line 461, what does 3mm of trans well insert mean? Is that a pore size? Please elaborate
Response. “Pore size” has been included.
- Line 425, TNF-a is repeated, please delete. Line 430, please replace “nN” with “n”
Response. The changes have been made. Thanks.
Reviewer 2 Report
In the proposed manuscript the author deeply investigated the involvement of TNFalpha in modulating LDL in an in vitro model of primary human aortic endothelial cells (pHAECs). Thorough a wide range of methodologies, the author demonstrated that TNFalpha plays a key role in LDL accumulation by enhancing LDL receptor functions. I have some comments and I suggest the following minor revisions:
- Please, provide in Fig.1 representative images of pHAECs showing morphology and level of confluency (staining with some specific endothelial cell markers such as von Willenbrand or CD31.
- Have these cells been previously demonstrated to be functional? Please, include in the manuscript references, if any available in literature.
- Regarding the concentration of TNFalpha used in Fig.1 (100ng/ml) how has this concentration been chosen? Was it extrapolated from previous evidence or from literature? Please, include this information
- Also, regarding Fig. 1, did the author perform a dose-response experiment? Is there a concentration where TNFalpha has inhibitor effects?
- In Fig.6 please specify the statistic (the n of the experiment)
- In Fig.6A the author needs to show GAPDH to ensure protein equal loading in the western blotting
- Regarding Fig.11 please add information about the composition of apical and basolateral medium, and details about how the transwell experiment has been performed (how many cells have been coated in the transwell insert? Was the insert pre-coated with any protein? Did the different media affect somehow pHAECs, in terms of morphology or functionality or cell adhesion?
Round 2
Reviewer 1 Report
Thank you for addressing all of my comments. The revised manuscripts look better.Author Response
Additional information has been included under Transwell Insert Experiments section of the Materials and Methods. All changes are indicated with left vertical bars.